# Assessment of Influenza Vaccine Uptake According to the Presence of a Chronic Disease

**DOI:** 10.3390/vaccines11050938

**Published:** 2023-05-04

**Authors:** Ibrahim M. Gosadi, Basem Zogel, Samar Alfaifi, Faisal Abusageah, Khalid M. Hakami, Taif Zogel, Tahani Altubayqi, Afnan Darsi, Ghaida Almuhaysin, Ali Ghalibi, Mohammed Alhazmi, Khulud Mahla

**Affiliations:** Department of Family and Community Medicine, Faculty of Medicine, Jazan University, Jazan 82621, Saudi Arabia

**Keywords:** influenza, vaccine, chronic diseases, uptake, Saudi Arabia

## Abstract

Patients diagnosed with a chronic disease are at higher risk of influenza complications. This investigation aimed to estimate influenza vaccine uptake rates among healthy subjects and patients with chronic diseases, and to identify barriers to and facilitators of its uptake. This study was a cross-sectional investigation that targeted the general population in the Jazan region of Saudi Arabia. Data were collected via online platforms between October and November 2022. Data were collected via a self-administered questionnaire that assessed the demographics, influenza vaccine uptake, and factors associated with the uptake. A chi-squared test was used to investigate factors associated with the uptake of the influenza vaccine. A total of 825 adult subjects participated in the current investigation. The proportion of male participants was higher (61%) compared to females (38%). The mean age of the participants was 36, with a standard deviation of 10.5. Nearly 30% of the sample reported being diagnosed with a chronic disease. Among the recruited sample, 576 (69.8%) reported having ever received the influenza vaccine, and only 222 participants (27%) indicated that they receive the influenza vaccination annually. Only history of being diagnosed with a chronic disease was statistically associated with the history of ever receiving the influenza vaccine (*p* < 0.001). Among the 249 participants with a chronic disease, only 103 (41.4%) ever received the influenza vaccine and only 43 (17.3%) of them received the vaccine annually. The main barrier to the uptake was fear of side effects due to the vaccination. A minority of the participants indicated that they were motivated by a healthcare worker to have the vaccine. This suggests an area for further investigation to assess the involvement of healthcare workers in motivating their patients with chronic diseases to have the vaccine.

## 1. Introduction

Influenza is a common seasonal acute respiratory infection caused by several influenza viruses. It is characterized by acute fever, cough, headache, muscle and joint pain, malaise, sore throat, and rhinorrhea that may recover spontaneously in most cases. However, some might suffer from complications that require further medical care. According to the World Health Organization (WHO), annual influenza epidemics can result in five million severe cases of influenza, especially among high-risk groups, of whom 650,000 may die due to respiratory complications [1].

The severity of the influenza infection is variable according to patients’ characteristics. It can vary according to the age group and can be augmented by health conditions and the presence of chronic comorbidity. Children under five years old and elderly people have been indicated as being at higher risk of developing severe complications related to influenza infection in comparison to other age groups. Pregnant women have also been reported to be at risk of developing severe influenza infection. Additionally, patients who are suffering from chronic diseases and those who are suffering from suppression of their immune system are at higher risk of being exposed to severe consequences of influenza infection [2].

Influenza infection is caused by different types of seasonal viruses, namely, A, B, C, and D. These viruses have rapid transmission capabilities and can be easily transmitted in crowded areas [1]. Influenza has been reported to be transmitted via several routes, including indirectly via contaminated surfaces and by contaminated hands, and directly by droplets or airborne transmission [3]. Practicing appropriate respiratory hygiene when coughing and sneezing, and regular hand washing have been suggested as preventive measures against the transmission of influenza. Nonetheless, the influenza vaccine is indicated as the most important measure for the prevention of influenza infection [2].

The influenza vaccination is recommended to be administered annually due to the probability of waning immunity over a year, and due to the constantly evolving nature of the influenza viruses. The WHO Global Influenza Surveillance and Response System conducts a continuous assessment of influenza viruses circulating in humans and provides updates concerning influenza vaccine compositions [1]. According to the American Center for Disease Control and Prevention, influenza vaccination is recommended to be administered annually prior to the expected onset of influenza outbreaks, thus strengthening immunity against the disease [2].

Annual vaccination of influenza is highly recommended for those who are vulnerable to severe complications of the infection. In Saudi Arabia, The Ministry of Health offers a free trivalent influenza vaccine (against A, B, and C types) on an annual basis, provides assistance concerning access to the vaccine, and promotes the uptake of the vaccine in preparation for the expected epidemics during the winter season [4]. According to the most recent monthly influenza updates of the East Mediterranean region, among the 13,209 tested cases from Saudi Arabia, a total of 1637 cases of influenza were detected during the analytical period of week 27, 2022 and week 8, 2023. Among the 1637 identified cases of influenza, 57% of the cases were flu B types and the remaining were flu A type [5].

Uptake of the vaccine among the general population has been reported to be affected by several determinants. Furthermore, though with limited evidence, uptake of the vaccine among high-risk groups in Saudi Arabia appears to be suboptimal. Nonetheless, a gap in knowledge exists concerning uptake rates of the vaccine among patients diagnosed with different chronic diseases, such as patients diagnosed with metabolic diseases, cancer, and blood disorders. This investigation aims to estimate influenza vaccine uptake rates among a sample from the Jazan region, including healthy subjects and patients diagnosed with multiple chronic diseases, and to identify barriers to and facilitators of the uptake among healthy subjects and patients diagnosed with chronic diseases.

## 2. Materials and Methods

### 2.1. Study Design and Settings

This study was a cross-sectional investigation with case–control analysis that targeted the general population in the Jazan region of Saudi Arabia. The required sample included patients diagnosed with various chronic diseases in addition to healthy adult subjects. Data were collected via online platforms between October and November 2022. The study was performed in accordance with the Declaration of Helsinki. Participation was declared to be voluntary and anonymous. Informed consent was obtained from the study participants prior to the study commencement. Ethical approval to perform the study was secured from the Jazan University Standing Committee for Scientific Research (approval number REC-44/02/298, dated 15 September 2022).

### 2.2. Data Collection Tool

Data collection was performed via a self-administered questionnaire (Appendix A), which was prepared after consulting the relevant literature that assessed the prevalence of influenza vaccine uptake and factors that are considered motivators or barriers to the uptake [6,7,8,9,10]. The developed questionnaire involved items measuring the demographics of the participants, history of diagnosis with a chronic disease, uptake of the influenza vaccine, and factors associated with the uptake of the vaccine. Furthermore, the questionnaire was piloted using a sample of five male and five female participants to assess its face validity, the clarity of the utilized items, and the time required to complete it.

### 2.3. Data Collection Process

The data collection tool was converted into electronic format to facilitate its distribution and recruitment of the target population. A web link was generated and distributed via social media groups to reach adult subjects in the Jazan region. All adult patients living in the Jazan region were included in the investigation. A screening question was listed at the beginning of the online questionnaire to enquire about the governorate of residence in the Jazan region. Those who indicated that they were not a resident of Jazan were excluded from the study.

This investigation utilized a non-probability, convenient, non-random sampling method to reach the required sample size. Studies conducted in Saudi Arabia that assessed the prevalence of influenza vaccine uptake among patients diagnosed with chronic diseases are limited to studies that assessed the uptake among patients diagnosed with type 2 diabetes. The current evidence suggests that the uptake of the vaccine among patients diagnosed with type 2 diabetes in Saudi Arabia varies between 61% and 43%. A mid-point of the uptake prevalence of 52% was used to estimate the required sample size to investigate the uptake of the vaccine among patients diagnosed with chronic diseases in the Jazan region. Using the StatCal function of the Epi Info software, a prevalence of 52% was used to estimate the sample, revealing a required sample of 383, assuming a 5% margin of error and 95% confidence interval level. The sample was doubled to account for the requirement of healthy subjects and to allow the comparison of the uptake among the two groups.

### 2.4. Data Analysis

Data analysis was conducted using the Statistical Package for Social Sciences v.25. Binary and categorical variables were summarized using frequencies and proportions. After the assessment of the distribution of continuous data and the detection of non-normal distribution, medians and the interquartile range were used to summarize continuous data. A chi-squared test was used to investigate factors associated with the influenza vaccine uptake among the recruited sample. To enable comparisons between the measured demographic variables and to avoid of having empty cells due to low numbers, marital status was grouped as being married or not, the income levels were grouped according to receiving a monthly income of 10,000 Saudi Arabian Riyal (SAR), education levels were grouped according to having a university education or not, the occupational status was grouped according to being employed or not, and smoking status was grouped as having ever smoked or not. A *p* value of 0.05 was designated as a value indicating the presence of a statistically significant difference.

## 3. Results

A total of 825 adult subjects from Jazan participated in the current investigation. The demographic characteristics of the participants are displayed in Table 1. The proportion of male participants was higher (61%) in comparison to females (38%), and the majority were Saudis (97%). The mean age of the participants was 36, with a standard deviation of 10.5. More than half of the participants belonged to rural areas from Jazan (55%), and the majority of the participants were married (68.6%). The majority of the participants had a monthly income of 5000 SAR or more and were educated to a tertiary level (71%). More than half of the sample were government employees, and, finally, the majority reported never having smoked (69.7%).

Table 2 displays the reported history of diagnosed chronic diseases among the recruited sample. Nearly 30% of the sample reported being diagnosed with a chronic disease, of which the most frequently diagnosed chronic conditions were hypertension (11%), diabetes (10.7%), and asthma (6.2%). Finally, 579 participants indicated not being diagnosed with any chronic condition.

Among the recruited sample, 576 (69.8%) reported having received the influenza vaccine and only 222 participants (27%) indicated that they receive the influenza vaccination on an annual basis. Table 3 displays influenza vaccine receipt motivators among the recruited sample. Among the participants who reported having ever received the vaccine, the most frequently reported motivator was believing that having the vaccine was generally important (37%), and specifically important as a protection against the incidence of complications of influenza (37.9%) or important for reducing the risk of infection (29.6%).

Table 4 summarizes factors that were indicated as barriers against having the influenza vaccine among the recruited sample. The most frequently reported reason was being unaware that it was advised to have the vaccine (12.2%), followed by not being informed by the treating physician to have it (11.6%), and fearing the incidence of vaccine-related side effects (10.8%). When the participants were asked whether they think the COVID-19 pandemic influenced them having the influenza vaccination, 34% of the sample reported that they think they were more committed to the vaccination before the pandemic, and 25.7% believe it became more difficult to receive the influenza vaccine after the pandemic. Nearly one-fifth of the sample reported their belief that the COVID-19 vaccine can either replace or conflict with the influenza vaccine.

Table 5 displays the association analysis between the history of receiving the influenza vaccination according to the measured sample characteristics. Only a history of being diagnosed with a chronic disease was statistically associated with the history of ever receiving the influenza vaccine (*p* < 0.001). Among the 249 participants who reported being diagnosed with a chronic disease, only 103 (41.4%) had ever received the influenza vaccine. The analysis was further continued to assess the receipt of the vaccine on an annual basis, according to being diagnosed with a chronic disease where a statistically significant difference was detected, indicating that only 43 (17.3%) participants with chronic diseases received the vaccine annually (*p* < 0.001).

## 4. Discussion

This investigation was a cross-sectional study performed in Jazan, Saudi Arabia to assess the uptake of influenza vaccination and factors associated with the uptake, especially according to the history of being diagnosed with a chronic disease. Nearly 70% of the total sample indicated they had received the vaccine, and only 27% reported annual receipt of the vaccine. When the analysis was performed to assess the receipt of the vaccine according to being diagnosed with a chronic disease, less than half of the participants with a chronic disease indicated having ever received the vaccine, while only 17.3% with chronic diseases reported annual vaccination with the influenza vaccine. The main motivators for having the vaccine were awareness of the seriousness of the disease and its complications and the importance of the vaccine in the prevention of the infection and the incidence of complications. The main barriers to vaccination among the participants were not being aware to have the vaccine, not being motivated to take the vaccine by their physicians, and fear of side effects.

The findings of this study can be compared to similar investigations conducted in Saudi Arabia that targeted the receipt of the influenza vaccination recommended for vulnerable groups. Studies that assessed the receipt of the influenza vaccination among patients diagnosed with chronic diseases in Saudi Arabia are currently limited. In addition to patients with chronic diseases, receipt of the vaccine is recommended for pregnant women, children aged between 6 months and 5 years, people above 65 years of age, pregnant women, and healthcare workers [1].

Several investigations have been conducted in Saudi Arabia to measure the uptake of influenza vaccines among different populations. A recent online investigation to measure willingness to take up COVID-19 and influenza vaccines among a sample of 1539 residents of Saudi Arabia, found that nearly 60% of the sample were willing to have the COVID-19 vaccine in comparison to only 31.7% who were willing to have the influenza vaccine [11]. The variation between willingness to have the COVID-19 vaccine and the influenza vaccine in Saudi Arabia can be partially explained by the fact that COVID-19 vaccination was mandated. Additionally, among a sample of 663 healthcare workers from Saudi Arabia, it was reported that 44% of them had the influenza vaccine in 2015 while the main reason for refusing to have the vaccine was a belief that the influenza vaccination was not necessary [12]. Furthermore, in a study that assessed the uptake of the influenza vaccine among a sample of 421 medical students from Riyadh, Saudi Arabia, it was concluded that only 20.7% had the vaccine in 2015 [13].

There are several additional investigations conducted in Saudi Arabia to measure influenza vaccine uptake among high-risk groups. Among a sample of 410 pregnant women from the eastern region of Saudi Arabia, less than 20% said they had received the vaccine [14]. Similarly, in a study that assessed the uptake of the vaccine among children in Saudi Arabia by asking 399 parents, only 37.6% of the children were indicated to have ever received the vaccine [15]. Finally, among the studies that assessed influenza vaccination among patients affected with type 2 diabetes, the uptake of the vaccine was reported to reach 61% in Abha [7], 47.8% in Riyadh [8], and 43.5% in Taif [16].

Table 6 displays a comparison between the findings of the current study and the findings of other investigations that targeted populations who were recommended to have the vaccine. It can be noted that the vaccine uptake varies according to the population, reaching the highest rates among a sample of healthcare workers from the Makkah region (88.3% in 2015) [17]. However, another online survey targeting healthcare workers indicated a lower vaccine uptake (44.1% in 2015) [12], which may suggest a variation in the attitude of healthcare workers toward the uptake of the influenza vaccine though it is not conclusive, due to variation in the methodology.

Among the studies that assessed the uptake of the vaccine among patients diagnosed with a chronic disease (namely diabetes), it can be noted that uptake rates are similar to our findings. In a study conducted in Abha, the uptake rate of the vaccine was 61%[7], while that among a sample of patients with diabetes from Riyadh was 47.8% [8]. These uptake rates are higher than that detected in our investigation (41%). Nonetheless, it can be noted that the uptake rate of the vaccine among other vulnerable groups, specifically pregnant women and children, is much lower in comparison to vaccine uptake among patients with chronic diseases and healthcare workers. The uptake among a sample of pregnant women was nearly 20%, as revealed by two surveys conducted in the eastern region of Saudi Arabia [14,18], while the uptake among children, as reported by their parents, is only 15% [15].

Table 6 provides a summary of the main motivators and barriers detected concerning the uptake of the influenza vaccine among vulnerable groups. Among the identified populations, being aware of influenza, the importance of the vaccine, and receiving advice from healthcare workers were the main motivators for having the vaccine. Fear of side effects of the vaccine was the main barrier against having it and frequently recurred as a barrier across several investigations. Additionally, as identified in a study targeting parents from Saudi Arabia, believing that natural immunity acquired from the infection was sufficient protection for the prevention of influenza was identified among the sample of parents, suggesting a misconception amongst parents concerning the importance of annual influenza vaccinations [15].

Our current investigation did not detect a statistically significant difference concerning vaccine uptake according to different age groups, though higher uptake was identified among participants younger than 35 years. In a study that surveyed 790 participants, who are aged 15 or older, from Saudi Arabia, it was concluded that those who are younger than 24 years reported the highest uptake rates in comparison to other age groups [19]. Furthermore, in a study that assessed influenza vaccine uptake among people aged 65 years and older, it was concluded that less than half of the 496 recruited participants reported ever receiving the vaccine [20]. These findings may suggest lower uptake rates of the vaccine among older individuals in comparison to younger ones in Saudi Arabia.

Comparing the findings of the current investigation to international investigations assessing the uptake of the vaccine among patients with chronic diseases revealed similar findings. A large US study, that recruited a sample of 36,811 patients with chronic obstructive pulmonary disease, in 2012 indicated that 48.5% of the patients had received the flu vaccine [21]. Furthermore, a Polish study that recruited a sample of 219 indicated that influenza vaccine uptake was only 26.5% by the end of the 2012–2013 season [22]. In a Chinese study that recruited a sample of 1914 patients with diabetes, the uptake rate of the influenza vaccine in the 2016–2017 season was only 7.84%, where residence, history of comorbidities, perceived benefits of the vaccine, and having free access to the vaccine were the main motivators for the uptake [23]. The findings of our current study and similar international investigation indicate the suboptimal vaccination rates among patients with chronic diseases.

The participants in the current investigation were asked whether the COVID-19 pandemic might have impacted their acceptance of the influenza vaccine. Nearly one-third of the sample indicated that they were more committed to receiving the vaccine, and 25% of the sample reported that it became more difficult to obtain the vaccine after the pandemic. Similarly, nearly one-fifth indicated that they believed there was a conflict between the COVID-19 vaccine and the influenza vaccine. Although our study did not measure the trend of the uptake of the influenza vaccine before and after the pandemic in the studied community, a reduction in the uptake of the influenza vaccine can be postulated. Nonetheless, this notion conflicts with the findings of a systematic review that indicated the intention to uptake the influenza vaccine was increased after the pandemic [24].

This study has multiple areas of strength and limitation. The main area of strength was related to its ability to reach a sample with different comorbidities and thus being able to assess the receipt of the influenza vaccine among one of the main vulnerable groups. Nonetheless, using an online approach introduces selection bias, as illiteratesubjects, elderly subjects, and those with limited access to online social platforms might be less likely to participate. Additionally, this study relied on the participants to indicate whether they were diagnosed with a chronic condition or not, thus, subjecting the study to possible reporting bias.

## 5. Conclusions

This study identified a suboptimal uptake of the influenza vaccine among patients diagnosed with chronic diseases in the Jazan region, even when compared to the uptake among healthy subjects. The main reasons associated with the low uptake were related to fear of the incidence of side effects due to the vaccine. A minority of the participants indicated that they were motivated by a healthcare worker to have the vaccine. This suggests an area for further investigation to assess the involvement of healthcare workers in motivating their patients with chronic diseases to have the vaccine.

## Figures and Tables

**Table 1 vaccines-11-00938-t001:** Demographic characteristics of 825 adult subjects from Jazan, Saudi Arabia.

Variable	Frequency [Proportion]
**Gender**	
Male	506 [61.3%]
Female	319 [38.7%]
Nationality	
Saudi	802 [97.2%]
Non-Saudi	23 [2.8]
Living place	
Urban	366 [44.4%]
Rural	459 [55.6%]
Marital status	
Married	566 [68.6%]
Widow	11 [1.3%]
Divorced	14 [1.7%]
Single	234 [28.4]
Monthly income (SAR)	
<5000	223 [27%]
5000–<10,000	179 [21.7%]
10,000–15,000	271 [32.8%]
>15,000	152 [18.4%]
Educational level	
Primary school	23 [2.8%]
Intermediate school	12 [1.5%]
High school	205 [24. 8%]
Bachelor’s degree or diploma	554 [67.2%]
Postgraduate	31 [3.8%]
Occupation	
Government Employee	426 [51.6%]
private Employee	75 [9.1%]
Business owner	10 [1.2%]
Housewife	64 [7.8%]
Retired	47 [5.7%]
Student	130 [15.8%]
Unemployed	73 [8.8%]
Smoking status	
Current smoking	160 [19.4%]
Ex-smoker	90 [10.9%]
Never	575 [69.7%]

**Table 2 vaccines-11-00938-t002:** Reported history of chronic conditions among 825 adult subjects from Jazan, Saudi Arabia.

Variables	Frequency [Proportion]
Reported diagnosed chronic disease *	
Hypertension	91 [11%]
Diabetes	88 [10.7%]
Asthma	51 [6.2%]
Sickle cell disease	19 [2.3%]
Obesity	15 [1.8%]
Cardiovascular disease	8 [1%]
Rheumatic disease	7 [0.8%]
Endocrine disorders	6 [0.7%]
Immunodeficiency	5 [0.6%]
Bronchitis	4 [0.5%]
Cancer	3 [0.4%]
Chronic kidney disease	3 [0.4%]
Other	20 [2.4%]
History of Co-morbidity	
Participants diagnosed with three diseases	11 [1.3%]
Participants diagnosed with two diaeseases	45 [5.4%]
Participants with one disease	193 [23.4%]

* Participants were able to select more than one answer.

**Table 3 vaccines-11-00938-t003:** Influenza vaccine receipt motivators among 576 adults from Jazan, Saudi Arabia.

Statement *	Frequency [Proportion]
It is important to take the vaccine	213 [37%]
Taking the vaccine may reduce the incidence of complications	218 [37.9%]
The influenza vaccine protects against influenza	170 [29.6%]
I took it to perform Hajj or Umrah	85 [14.8%]
There was no particular reason for me to take the vaccine	70 [12.2%]
I took it after getting advice from my healthcare provider	67 [11.7%]
I took it after I got advice from a friend/relative	50 [8.7%]
I took it after attending an awareness campaign	47 [8.2%]
I took it because it was required by my employer	31 [5.4%]
Others	6 [1%]

* Participants were able to select more than one statement.

**Table 4 vaccines-11-00938-t004:** Statements indicating the presence of influenza vaccination receipt barriers among 825 adult subjects from Jazan, Saudi Arabia.

Statements *	Frequency [Proportions]
I did not know I had to get the influenza vaccine	101 [12.2%]
My physician did not state that I had to get vaccinated	96 [11.6%]
I am afraid of the side effects that may appear after vaccination.	89 [10.8%]
I do not know when and where to vaccinate	57 [6.9%]
I do not think the vaccine gives comprehensive protection against influenza	55 [6.7%]
I do not need to get vaccinated because I am healthy	52 [6.3%]
Influenza is not a serious illness	41 [5%]
I do not have time to get vaccinated	31 [3.8%]
I do not think the vaccine is safe	26 [3.2%]
I am afraid of needle pricks	17 [2.1%]
I cannot find the vaccine in PHC center	16 [1.9%]
I have had a bad previous experience with the flu vaccine	13 [1.6%]
In general, I am against vaccination.	11 [1.3%]
I have mobility difficulties and I can only go to the doctor, if someone takes me	3 [0.4%]
No barriers reported	476 [57.7%]
Statements indicating interference between COVID-19 vaccine and the influenza vaccine.	
I think I was relatively more committed to getting the influenza vaccine before COVID-19 pandemic in comparison to the period of COVID-19 pandemic	281 [34%]
I think it has become difficult to obtain the influenza vaccine after the advent of the COVID-19 pandemic.	212 [25.7%]
I think that the COVID-19 vaccine replaces the influenza vaccine.	181 [22%]
I think the influenza vaccine conflicts with the COVID-19 vaccine.	169 [20.4%]

* Participants were able to select more than one statement.

**Table 5 vaccines-11-00938-t005:** Factors associated with the receipt of influenza vaccination among 825 adult subjects from Jazan, Saudi Arabia.

	History of Ever Receipt of the Influenza Vaccine		Total	*p* Value
	No	Yes		
Age				0.196
<35 years	107 [27.9%]	277 [72.1%]	384 [100%]	
≥35 years	142 [32.2%]	299 [67.8%]	441 [100%]	
Marital status				0.142
Married	180 [31.8%]	386 [68.2%]	566 [100%]	
Not Married	69 [26.6%]	190 [73.4%]	259 [100%]	
Monthly income				0.879
More than 10,000 SAR	129 [30.5%]	294 [69.5]	423 [100%]	
10,000 SAR or less	120 [29.9%]	282 [70.1%]	402 [100%]	
Education				0.560
Less than University education	76 [31.7%]	164 [68.3%]	240 [100%]	
University education	173 [29.6%]	412 [70.4%]	585 [100%]	
Occupation				0.185
Employee or business owner	163 [31.9%]	348 [68.1%]	511 [100%]	
Not employee	86 [27.4%]	228 [72.6%]	314 [100%]	
Smoking				0.158
Ever smoker	84 [33.6%]	166 [66.4%]	250 [100%]	
Never smoker	165 [28.7%]	410 [71.3%]	575 [100%]	
Living place				0.285
Rural	103 [28.1%]	263 [71.9%]	366 [100%]	
Urban	146 [31.8%]	313 [68.2%]	459 [100%]	
Diagnosis of a chronic disease				<0.001
Yes	103 [41.4%]	146 [58.6%]	249 [100%]	
No	146 [25.3%]	430 [74.7%]	576 [100%]	

**Table 6 vaccines-11-00938-t006:** Comparison of studies conducted in Saudi Arabia to assess uptake of the influenza vaccination among groups highly recommended to receive the vaccine.

Year of the Study	Location	Sample Size	Population	Vaccine Uptake	Main Motivators	Main Barriers	Reference
Current study	Jazan, Saudi Arabia	825	70% healthy 30% with chronic diseases	25% among the healthy 41% among those with chronic conditions	Awareness about seriousness of influenza	Lack of motivation and fear of side effects	Current study
2019	Riyadh, Saudi Arabia	360	type 2 diabetes	47.8%	Recognizing the importance of the vaccine	Age, marital status, and education level	[8]
2017–2018	Abha, Saudi Arabia	353	type 2 diabetes mellitus (T2DM) patients	61%	Healthcare givers’ advice	Fear of side effects	[7]
2017–2018	Dammam and Al-Khobar, Saudi Arabia	410	pregnant women	19.8%	Healthcare givers’ advice	Fear of side effects	[14]
2019–2020	Al-Ahsa, Saudi Arabia	404	pregnant women	20.3%	Not reported	Fear of side effects	[18]
2019	Qassim, Saudi Arabia	399	Saudi parents	85% of the parents reported not vaccinating their children	knowledge and attitude about influenza	Believing that natural immunity is sufficient for influenza prevention	[15]
2015	Makkah, Saudi Arabia	447	healthcare workers	88.3%	Self-protection	Believing that the vaccine causes influenza	[17]
2015	Online settings, Saudi Arabia	633	healthcare workers	44.1%	Believing that the vaccine is effective in preventing the infection	Belief that the vaccine was not necessary	[12]

## Data Availability

The data presented in this study are available on request from the corresponding author.

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
