# Peer review of "Assessment of Influenza Vaccine Uptake According to the Presence of a Chronic Disease"

_vaccines, 2023, doi:10.3390/vaccines11050938_

Round 1

Reviewer 1 Report

 Dear Editor and Authors,

 I have read your submission on influenza vaccine uptake rates and barriers/facilitators among healthy individuals and those with chronic diseases in the Jazan region of Saudi Arabia with great interest. While the study provides valuable insights into the barriers/facilitators to flu vaccination, I have some concerns about the methodology and findings.

Firstly, the study relies on self-reported medical histories of participants without reference to hospital records. This may introduce bias into the results. It would be helpful if the methods section clarified how the population was selected based on medical history records, or if not, acknowledged this limitation in the discussion.

Secondly, the study does not explain why individuals with chronic diseases are more motivated to receive the flu vaccine compared to other vaccines like Hepatitis. It is unclear whether healthcare workers only motivate their patients to get the flu vaccine or if they also recommend other vaccines. It would be valuable if the authors could provide information related to vaccine hesitancy among chronic people compared to healthy individuals who did not choose to get vaccinated. This could open up another discussion about how having a chronic disease influences vaccine acceptance compared to those who identify themselves as healthy.

Lastly, I have some minor comments on the manuscript, which I have detailed in the returned document.

Thank you for your attention to these concerns if you decide to reply 

Sincerely,

Author Response

General comments

Dear Editor and Authors,

Comment: I have read your submission on influenza vaccine uptake rates and barriers/facilitators among healthy individuals and those with chronic diseases in the Jazan region of Saudi Arabia with great interest. While the study provides valuable insights into the barriers/facilitators to flu vaccination, I have some concerns about the methodology and findings.

Response: The authors of the manuscript appreciate the comment of the reviewer. A thorough  revision has been applied to all sections of the manuscript to enhance the writing quality of the manuscript. A change track function has been activated in the revised manuscript to indicate areas of modifications.

Comment: Firstly, the study relies on self-reported medical histories of participants without reference to hospital records. This may introduce bias into the results. It would be helpful if the methods section clarified how the population was selected based on medical history records, or if not, acknowledged this limitation in the discussion.

Response: The authors of the manuscript agree with the comment of the reviewer where the questionnaire-based assessment may subject the study to measurement bias. This investigation did not target medical records of the patients, yet, it enabled those who are interested to participate to voluntarily complete the questionnaire. Though measurement bias cannot be entirely avoided in this instance, the questionnaire involved an item assessing whether the participates were diagnosed by a medical professional with a chronic disease . This notion is now added to the revised manuscript to indicate this as a limitation as the following:

‘Additionally, this study relied on the participants to indicate whether they were diagnosed with a chronic condition or not, thus, subjecting the study to possible reporting bias’ 

Comment: Secondly, the study does not explain why individuals with chronic diseases are more motivated to receive the flu vaccine compared to other vaccines like Hepatitis. It is unclear whether healthcare workers only motivate their patients to get the flu vaccine or if they also recommend other vaccines.

Response:  The authors appreciate the comment of the reviewer. However, the current investigation did not compare motivation to receive the flu vaccine to other types of vaccines. The current investigation was aiming to estimate influenza vaccine uptake rates and compare the uptake according to the presence of a diagnosed chronic disease. Nonetheless, comparing the uptake according to the type of the vaccine can be an area for further investigation.

Comment: It would be valuable if the authors could provide information related to vaccine hesitancy among chronic people compared to healthy individuals who did not choose to get vaccinated. This could open up another discussion about how having a chronic disease influences vaccine acceptance compared to those who identify themselves as healthy.

Response: The authors of the manuscript value the comment of the reviewer. After careful and extensive review of the literature, it was noted that studies that compared vaccine hesitancy and vaccination acceptance as a general concept between individuals diagnosed with a chronic disease and healthy individuals are limited. A review titled ‘COVID-19 vaccine hesitancy and acceptance among the public in the Gulf Cooperation Council countries: A review of the literature’ [1] was identified and discussed vaccine hesitancy among different populations but did not contain comparison of vaccine hesitancy between those who are diagnosed with a chronic disease and healthy individuals. Furthermore, in another study that assessed knowledge, attitude and self-reported adherence concerning influenza vaccination among an Italian population was limited to 414 patients diagnosed with various chronic diseases but did not involve healthy individuals [2]. Finally, a Mexican study was identified to compare vaccine hesitancy between healthy individuals and adults with chronic diseases but was limited to COVID-19 vaccination hesitancy only and was not generalized to other vaccines such as influenza [3].  Therefore, it was difficult to identify studies that compared vaccine hesitancy (including hesitancy toward influenza vaccine) between patients diagnosed with chronic disease and healthy individuals which suggests the importance and novelty of our current investigation that was able to reach a sample of healthy individuals and patients diagnosed with a chronic disease to enable the assessment of influenza vaccine receipt according to the presence of a chronic disease.  

References:

  1. Alsalloum MA, Garwan YM, Jose J, Thabit AK, Baghdady N. COVID-19 vaccine hesitancy and acceptance among the public in the Gulf Cooperation Council countries: A review of the literature. Hum Vaccin Immunother. 2022 Nov 30;18(6):2091898. doi: 10.1080/21645515.2022.2091898. Epub 2022 Jun 29. PMID: 35767457; PMCID: PMC9746509.
  2. Napolitano F, Della Polla G, Capano MS, Augimeri M, Angelillo IF. Vaccinations and Chronic Diseases: Knowledge, Attitudes, and Self-Reported Adherence among Patients in Italy. Vaccines (Basel). 2020 Sep 25;8(4):560. doi: 10.3390/vaccines8040560. PMID: 32992864; PMCID: PMC7711873.
  3. González-Block MÁ, Gutiérrez-Calderón E, Sarti E. COVID-19 Vaccination Hesitancy in Mexico City among Healthy Adults and Adults with Chronic Diseases: A Survey of Complacency, Confidence, and Convenience Challenges in the Transition to Endemic Control. Vaccines (Basel). 2022 Nov 17;10(11):1944. doi: 10.3390/vaccines10111944. PMID: 36423039; PMCID: PMC9694314.

Comment: Lastly, I have some minor comments on the manuscript, which I have detailed in the returned document. Thank you for your attention to these concerns if you decide to reply 

Response: The authors appreciate the valuable comments of the reviewer. All requested modifications were applied to the revised manuscript to enhance the rewriting quality. All modifications applied to the manuscript as marked via the track changes function in the revised manuscript.

Minor comments

Abstract:

Comment: remove word *purpose*

Response: The word was removed as requested

Comment:: remove word *methods*

Response: The word was removed as requested.

Comment:: remove word *results*

Response: The word was removed as requested.

Comment:: please check word *recipt*

Response: The word was checked and it appears correct. No modification was applied in response to this comment.

Comment:: remove word *conclusions*

Response: The word was removed as requested.

Introduction

Comment:: I suggest remove this part from 49 to 65

Response: The authors of the manuscript highly appreciate the comment of the reviewer. After careful consideration, we believe that the two indicated paragraphs are describing important notions necessary for the readers to have clearer understanding of influenza and why annual vaccination is recommended, which is the main aim of the study. Therefore, we believe that the indicated paragraphs are necessary and were not removed.

Comment:: please remove or move to discussion with a summary from 73 to 85... and same as from 86 to 92

Response: The indicated sections have been moved from the introduction to the discussion as per the request of the reviewer. The modifications are marked by the track changes function in the revised manuscript.

Methodology

Comment: It`s not clear what sample is required. Authors need to clearly describe this sample here and one above

Response: The authors of the manuscript appreciate the comment of the reviewer. The sample in the current context refers to performing a number of observations via a measurement tool ( a questionnaire in the current epidemiological study) to measure the frequency and associated factors of a particular condition (uptake of the flu vaccine in the current investigation). We believe that the current description, either in the introduction and methodology, is suitable to describe sampling of a population in epidemiological designs and no further modifications were applied to the revised manuscript in response to this particular comment.

Comment:: move the dot after the refs.

Response: The dot has been moved as requested.

Comment: Authors need to provide that questionnaire survey template as a supporting document

Response: The authors agree to provide the questionnaire as supporting document.

Comment: what are the samples obtained from them?

Response: The authors of the manuscript appreciate the comment of the reviewer. The sample in the current context refers to performing a number of observations via a measurement tool ( a questionnaire in the current epidemiological study) to measure the frequency and associated factors of a particular condition (uptake of the flu vaccine in the current investigation). We believe that the current description, either in the introduction and methodology, is suitable to describe sampling a population in epidemiological designs and no further modifications were applied to the revised manuscript in response to this particular comment. We also confirm that no biological samples were taken from the participants in the current study. 

Comment:: please consider using population instead of 'sample' size or more relevant identifier for those groups as it can create confusion with biological samples belong to participants.

Response: The authors of the manuscript appreciate the comment of the reviewer. The sample in the current context refers to performing a number of observations via a measurement tool ( a questionnaire in the current epidemiological study) to measure the frequency and the associated factors of a particular condition (uptake of the flu vaccine in the current investigation). We believe that the current description, either in the introduction and methodology, is suitable to describe sampling a population in epidemiological designs and no further modifications were applied to the revised manuscript in response to this particular comment. We also confirm that no biological samples were taken from the participants in the current study.  Finally, using the term ‘population’  instead of ‘sample’ might not be correct as the performed inferential statistics are based on a selected sample (limited number of observations) from a particular population ( all people who meet the inclusion criteria of the current context).

Results:

Comment: The comment on variable: what was the avarege

Response: The authors appreciate the comment of the reviewer. The indicated table is displaying frequencies and proportion of either binary or categorical variables measured in the current study.

Comment: The comment on monthly income : Please specify in what currency?

Response: The currency is now added to the indicated location in table 1 as (SAR). We also confirm that the abbreviation was spelled out upon its first appearance in the methodology section.

Comment:  The comment: please include total number of participants in the end.

Response: The total number of the participants is indicated title of the table (825) and was not added to each row in the table to avoid redundancy.  

Comment: Authors need to summarize what participant had how many chronic condition, you can provide the detailed table as a supplement.

Response: The authors appreciate the comment of the reviewer. Nonetheless, it must be noted that the comment was not entirely clear to the authors. However, table 2 was describing nature of reported diagnosed condition where each participant was able to report any chronic condition the participant was diagnosed with. To provide more details about history of co-morbidity (to indicate that some patients were diagnosed with more than one condition simultaneously, new rows are now added to table two to illustrate those who had comorbidities where the maximum number of diagnosed chronic conditions of a single individuals in our study was 3 and the minimum number of diagnosed chronic condition was 1 condition). The modified section of table 2 are indicated by track change function in the revised manuscript.

Comment:  In addition, authors need to move that statement '*' out of the table as table legend as it`s not a part of the list.

Response: the statement has been removed from the table and added beneath the table as a legend.

Comment:  It`s also needed to be highlighted that how authors ensure the reliability of the statements about the diagnosis as it can cause biased results since they are self reported.

Response: The authors agree with the comment of the reviewer and the following statement was added to the limitation section to increase the scientific transparency of the manuscript:

‘Additionally, this study relied on the participants to indicate whether they were diagnosed with a chronic condition or not, thus, subjecting the study to possible reporting bias’ 

Comment:  The comment: what is sample used for here

Response: The sample here refers to the participants whom were approached and agreed to complete the questionnaire.

Comment: : can you please explain how these statements were identified and selected? I am curious to know what question was asked.

Response: Data collection was performed via a self-administered questionnaire, which was prepared after consulting the relevant literature that assessed the prevalence of influenza vaccine uptake and factors that are considered motivators or barriers to the uptake. The following references were consulted to identify and select the questions:

  1. Alnaheelah, I.M., et al., Influenza Vaccination in Type 2 Diabetes Patients: Coverage Status and Its Determinants in Southwestern Saudi Arabia. Int J Environ Res Public Health, 2018. 15(7).
  2. Almusalam, Y.A., M.K. Ghorab, and S.L. Alanezi, Prevalence of influenza and pneumococcal vaccine uptake in Saudi type 2 diabetic individuals. J Family Med Prim Care, 2019. 8(6): p. 2112-2119.
  3. Korkmaz, P., et al., Influenza vaccination prevalence among the elderly and individuals with chronic disease, and factors affecting vaccination uptake. Cent Eur J Public Health, 2019. 27(1): p. 44-49.
  4. Santos, A.J., et al., Beliefs and attitudes towards the influenza vaccine in high-risk individuals. Epidemiol Infect, 2017. 145(9): p. 1786-1796.
  5. Yan, S., et al., Barriers to influenza vaccination among different populations in Shanghai. Hum Vaccin Immunother, 2021. 17(5): p. 1403-1411.

Source of the items were described in the methodology section of the manuscript.

Comment: Since 825 people were surveyed, it would be expected to get 825 different answers. 

Response: The authors of the manuscript appreciate the comment of the reviewer. The participants were asked to select barriers that they believe are interfering with their receipt of the vaccine. As explained in the results section, more than half of the participants reported receipt of the influenza vaccine. Additionally, 476 participants reported no presence of any barrier against taking the vaccine among the 825 participants. To increase the reporting quality, the proportion of those who reported no presence of barriers is now added to table 4 in a new additional row.

Comment: Could you clarify if these statements were part of the questionnaire in your survey?

Response: Yes, we confirm that the statement displayed in the table are from the utilized questionnaire. The authors agree to attach the questionnaire as a supplement to enhance the reporting quality of the revised manuscript.

Comment: Additionally, I would appreciate it if you could share the survey template that was used for the participants to fill out, as a supplement.

Response: The authors agree to attach the questionnaire as a supplement to enhance the reporting quality of the revised manuscript.

Comment:: I invite authors that please revise your tables as it`s being difficult to follow your headings and subheadings. Please do the same for others tables that you have.

Response: We confirm that all tables have been revisited and edited to enable easier differentiation between heading and subheadings within all tables. However, since we are using official Microsoft Word template provided by MDPI, it was difficult to save some editing changes, such as font size and use of bold in headings and subheadings. We confirm that will work closely with the editorial office for any editing modifications needed to be applied on the manuscript.

Comment:: Were they significantly important based on statistical analysis?

Response: The identified barriers were most frequently reported barriers among the participants. However, better phrasing is now used to report the nature of the barriers as the following:

‘The main barriers to the vaccinations among the participants were not being aware to have the vaccine, not being motivated to take the vaccine by their physicians, and fear of side effects.’

The modified text is indicated by the Track change function in the revised manuscript.

Comment: please use numerical

Response: Numerical is now used in the indicated text .

Comment:: I am curious that is anyone reported or authors asked to questions related to vaccine hesitancy and hate for flu vaccines.

Response: The authors of the manuscript appreciate the comment of the reviewer. However, vaccine hesitancy was not a primary aim of the current investigation and specific assessment tools used to measure general vaccine hesitancy were not targeted in the current investigation.

Reviewer 2 Report

The article represents a partial analysis of the vaccination of the Saudi Arabian population to the influenza vaccine. The report is general; the cohort involves a group of mainly young volunteers, mostly men, with a subgroup of individuals with chronic diseases. The authors did not include epidemiological data on influenza infection in the country nor the possible vaccination rate at other ages with higher risks. The questionnaire should be included in the supplemental file. The authors should discuss the local epidemiological data prior to and after the COVID-19 waves that have affected the outspread of influenza infection and what should be the new guidelines.

The article is general, it requires editorial work before publication.

Author Response

Comment: The article represents a partial analysis of the vaccination of the Saudi Arabian population to the influenza vaccine. The report is general; the cohort involves a group of mainly young volunteers, mostly men, with a subgroup of individuals with chronic diseases. The authors did not include epidemiological data on influenza infection in the country nor the possible vaccination rate at other ages with higher risks.

Response: The authors of the manuscript value the comment of the reviewer. However, it must be noted that epidemiological data on influenza infection in Saudi Arabia are not periodically published nor publically available on national level despite the establishment of the  Influenza Surveillance in Saudi Arabia (2017) [1]. However, limited earlier reports were identified that studied the epidemiology of influenza A H1N1 in the country during the 2009 pandemic [2-4]. Additionally, these studies were limited to the pandemic and may not provide relevant and recent epidemiological data on influenza infection in the country, and therefore, were not included in the revised manuscript.  Nonetheless, consulting the Global Influenza Program provided some insight concerning the epidemiology of influenza infection in Saudi Arabia [5]. However, it must be noted that epidemiological data presented by the program via its periodic reports are not provided on country levels but rather on regional levels (including the East Mediterranean region which includes Saudi Arabia). Therefore, the following description was added to the introduction based on the periodic reports of the Global Influenza Program to indicate the epidemiology of influenza infection in Saudi Arabia:

‘According to the most recent influenza monthly updates of the East Mediterranean region, among the tested 13209 cases from Saudi Arabia,  a total of 1637 cases of influenza were detected during the analytical period of week 27, 2022 and week 8, 2023. Among the identified 1637 cases of influenza, 57% of the cases were flu B types and the remaining were flu A type. ‘

References:

  1. Saudi Ministry of Health. Influenza Surveillance in Saudi Arabia. Available from:[https://www.moh.gov.sa/CCC/healthp/regulations/Documents/ISSA%20Protocol.pdf]. Accessed on 23th of April 2023.
  2. AlMazroa MA, Memish ZA, AlWadey AM. Pandemic influenza A (H1N1) in Saudi Arabia: description of the first one hundred cases. Ann Saudi Med. 2010 Jan-Feb;30(1):11-4. doi: 10.4103/0256-4947.59366. PMID: 20103952; PMCID: PMC2850176.
  3. Herzallah HK, Bubshait SA, Antony AK, Al-Otaibi ST. Incidence of influenza A H1N1 2009 infection in Eastern Saudi Arabian hospitals. Saudi Med J. 2011 Jun;32(6):598-602. PMID: 21666942.
  4. Agha A, Alrawi A, Munayco CV, Alayed MS, Al-Hakami M, Korairi H, Bella A. Characteristics of Patients Hospitalized with 2009 H1N1 Influenza in a Tertiary Care Hospital in Southern Saudi Arabia. Mediterr J Hematol Infect Dis. 2012;4(1):e2012002. doi: 10.4084/MJHID.2012.002. Epub 2012 Jan 6. PMID: 22348184; PMCID: PMC3279317.
  5. World Health Organization. The Global Influenza Program. Available from:[ https://www.who.int/teams/global-influenza-programme]. Accessed on 26th of April 2023.

The following texts discuss  possible vaccination rates at other ages with higher risks while displaying findings of the current investigation and comparing it to similar local literature: .

‘Table 6 displays a comparison between the findings of the current study and the findings of other investigations that targeted populations who were recommended to have the vaccine. It can be noted that the vaccine uptake varies according to the population, reaching the highest rates among a sample of healthcare workers from the Makkah region (88.3% in 2015) [1]. However, another online survey targeting healthcare workers indicated a lower vaccine uptake (44.1% in 2015) [2], which may suggest a variation in the attitude of healthcare workers toward the uptake of the influenza vaccine though is not conclusive due to variation in the methodology.

Among the studies that assessed the uptake of the vaccine among patients diagnosed with a chronic disease (namely diabetes), it can be noted that uptake rates are similar to our findings. In a study conducted in Abha, the uptake rate of the vaccine was 61%[3], while that among a sample of patients with diabetes from Riyadh was 47.8% [4]. These uptake rates are higher than that detected in our investigation (41%). Nonetheless, it can be noted that the uptake rate of the vaccine among other vulnerable groups, specifically pregnant women and children, is much lower in comparison to vaccine uptake among patients with chronic diseases and healthcare workers. The uptake among a sample of pregnant women was nearly 20%, as revealed by two surveys conducted in the Eastern region of Saudi Arabia [5, 6], while the uptake among children as reported by their parents is only 15% [7].

Table 6 provides a summary of the main motivators and barriers detected concerning the uptake of the influenza vaccine among vulnerable groups. Among the identified populations, being aware of influenza, the importance of the vaccine, and receiving advice from healthcare workers were the main motivators for having the vaccine. Fear of side effects of the vaccine was the main barrier against having it and frequently recurred as a barrier across several investigations. Additionally, as identified in a study targeting parents from Saudi Arabia, believing that natural immunity acquired from the infection was sufficient protection for the prevention of influenza was identified among the sample of parents, suggesting a misconception amongst parents concerning the importance of annual influenza vaccinations [7].

Our current investigation did not detect a statistically significance difference concerning vaccine uptake according to different age groups though higher uptake was identified among participants younger than 35 years. In a study that surveyed 790 participants whom are aged 15 or more from Saudi Arabia, it was concluded that those whom are younger than 24 years reported highest uptake rates in comparison to other age groups [8]. Furthermore, in a study that assessed influenza vaccine uptake among people aged 65 years and older, it was concluded that less half of the 496 recruited participants reported ever receipt of the vaccine [9]. These findings may suggests lower uptake rates of the vaccine among older individuals in comparison to younger ones in Saudi Arabia. ‘

Table 6: Comparison of studies conducted in Saudi Arabia to assess uptake of the influenza vaccination among groups highly recommended to the vaccine: 

Year of the study

Location

Sample size

Population

Vaccine uptake

Main motivators

Main barriers

Reference 

Current study

Jazan, Saudi Arabia

825

70% health

30% with chronic diseases

25% among the healthy

41% among those with chronic conditions

Awareness about seriousness of the influenza

Lack of motivation and fear of side effects

Current study

2019

Riyadh, Saudi Arabia

360

type 2 diabetes

47.8%

Recognizing the importance of the vaccine

Age, marital status, and education level

[4]

2017-2018

Abha, Saudi Arabia

353

type 2 diabetes mellitus (T2DM) patients

61%

Healthcare givers’ advice

Fear of side effects 

[3]

2017-2018

Dammam and Al-Khobar, Saudi Arabia

410

Pregnant women

19.8%

Healthcare givers’ advice

Fear of side effects 

[6]

2019-2020

Al-Ahsa, Saudi Arabia

404

pregnant women

20.3%

Not reported

Fear of side effects 

[5]

2019

Qassim, Saudi Arabia

399

Saudi parents

85% of the parents reported not vaccinating their children

knowledge and attitude about influenza

Believing that natural immunity is sufficient for influenza prevention

[7]

2015

Makkah, Saudi Arabia

447

healthcare workers

88.3% 

Self-protection

Believing that the vaccine causes influenza  

[1]

2015

Online settings, Saudi Arabia

633

healthcare workers

44.1%

Believing that the vaccine is effective in preventing the infection

Belief that the vaccine was not necessary

[2]

References:

  1. Haridi, H.K., et al., Influenza vaccine uptake, determinants, motivators, and barriers of the vaccine receipt among healthcare workers in a tertiary care hospital in Saudi Arabia. J Hosp Infect, 2017. 96(3): p. 268-275.
  2. Rabaan, A.A., et al., Influenza vaccine acceptance by healthcare workers in Saudi Arabia: A questionnaire-based analysis. Infez Med, 2020. 28(1): p. 70-77.
  3. Alnaheelah, I.M., et al., Influenza Vaccination in Type 2 Diabetes Patients: Coverage Status and Its Determinants in Southwestern Saudi Arabia. Int J Environ Res Public Health, 2018. 15(7).
  4. Almusalam, Y.A., M.K. Ghorab, and S.L. Alanezi, Prevalence of influenza and pneumococcal vaccine uptake in Saudi type 2 diabetic individuals. J Family Med Prim Care, 2019. 8(6): p. 2112-2119.
  5. Albattat, H.S., et al., Knowledge, attitude, and barriers of seasonal influenza vaccination among pregnant women visiting primary healthcare centers in Al-Ahsa, Saudi Arabia. 2019/2020. J Family Med Prim Care, 2021. 10(2): p. 783-790.
  6. AlMusailhi, S.A., N.M. AlShehri, and W.M. AlHarbi, Knowledge, utilization and barriers of pregnant women to influenza vaccine in primary health care centers in Dammam and Al Khobar, Saudi Arabia, 2017-2018. Int J Womens Health, 2019. 11: p. 207-211.
  7. Alolayan, A., et al., Seasonal Influenza Vaccination among Saudi Children: Parental Barriers and Willingness to Vaccinate Their Children. Int J Environ Res Public Health, 2019. 16(21).
  8. Sales, I.A., et al., Public Knowledge, Attitudes, and Practices toward Seasonal Influenza Vaccine in Saudi Arabia: A Cross-Sectional Study. Int J Environ Res Public Health, 2021. 18(2).
  9. Alotaibi, F.Y., et al., Influenza vaccine coverage, awareness, and beliefs regarding seasonal influenza vaccination among people aged 65 years and older in Central Saudi Arabia. Saudi Med J, 2019. 40(10): p. 1013-1018.

Comment: The questionnaire should be included in the supplemental file.

Response: The authors agree to attach the questionnaire as a supplement to enhance the reporting quality of the revised manuscript.

Comment: The authors should discuss the local epidemiological data prior to and after the COVID-19 waves that have affected the outspread of influenza infection and what should be the new guidelines.

  Response:

The authors of the manuscript value the comment of the reviewer. However, it must be noted that epidemiological data on influenza infection in Saudi Arabia are not periodically published nor publically available on national level. Additionally, the recent reports of the Global Influenza Program are reported on regional levels and are mainly limited to the number of tested cases per country and the identified types of influenza cases. After consulting the most recent influenza monthly updates of the East Mediterranean region, a reduction of number of tested specimens for influenza was noted during the period between 2020 and 2022. However, it is difficult from the available data to clearly indicate whether there was a difference of the outspread of influenza infection prior and after the COVID-19 pandemic. Having indicated the presence of limited local data concerning epidemiology of influenza on national level in Saudi Arabia, suggesting new guidelines based on the findings of the current investigation might be difficult. However, the following notion was added to the discussion suggesting potential impact of the pandemic:

‘The participants in the current investigation were asked whether the COVID-19 pandemic might have impacted their acceptance of the influenza vaccine. Nearly one-third of the sample indicated that they were more committed to receiving the vaccine, and 25% of the sample reported that it became more difficult to obtain the vaccine after the pandemic. Similarly, nearly one-fifth indicated they believed there was a conflict between the COVID-19 vaccine and the influenza vaccine. Although our study did not measure the trend of the uptake of the influenza vaccine before and after the pandemic in the studied community, a reduction in the uptake of the influenza vaccine can be postulated.’

Round 2

Reviewer 1 Report

Thanks for addressing my concerns through your revised draft and in your response letter. 

Reviewer 2 Report

The article was improved with the addition of the requested data and text. The discussion was enhanced as suggested. The article can be published

Minor details in the language should be corrected by the production department